# LBD: Decouple Relevance and Observation for Individual-Level Unbiased Learning to Rank

**Mouxiang Chen**[1,4*], **Chenghao Liu**[2*†], **Zemin Liu**[3], **Jianling Sun**[1,4†]

[1]Zhejiang University, [2]Salesforce Research Asia, [3] National University of Singapore,
[4]Alibaba-Zhejiang University Joint Institute of Frontier Technologies
{chenmx,sunjl}@zju.edu.cn, chenghao.liu@salesforce.com, zeminliu@nus.edu.sg

## Abstract

Using Unbiased Learning to Rank (ULTR) to train the ranking model with biased click logs has attracted increased research interest. The key idea is to explicitly model the user's observation behavior when building the ranker with a large number of click logs. Considering the simplicity, recent efforts are mainly based on the position bias hypothesis, in which the observation only depends on the position. However, this hypothesis does not hold in many scenarios due to the neglect of the distinct characteristics of individuals in the same position. On the other hand, directly modeling observation bias for each individual is quite challenging, since the effects of each individual's features on relevance and observation are entangled. It is difficult to ravel out this coupled effect and thus obtain a correct relevance model from click data. To address this issue, we first present the concept of coupling effect for individual-level ULTR. Then, we develop the novel Lipschitz and Bernoulli Decoupling (LBD) model to decouple the effects on relevance and observation at the individual level. We prove theoretically that our proposed method could recover the correct relevance order for the ranking objective. Empirical results on two LTR benchmark datasets show that the proposed model outperforms the state-of-the-art baselines and verify its effectiveness in debiasing data. Our codes are available at https://github.com/Keytoyze/Lipschitz-Bernoulli-Decoupling.

## 1  Introduction

Learning to Rank (LTR) has been widely used in modern information retrieval systems, which aims to learn a ranking model that sorts documents with their relevance. Recently, using implicit feedback (e.g. clicks) instead of relevance labels (typically obtained by human annotation) to train the ranking model has become popular since implicit feedback is an attractive proxy of relevance, which is cheap and relatively easy to obtain on a large scale [38]. However, they inherently contain a lot of bias from user behavior [28]. Using Unbiased Learning to Rank (ULTR) to remove these biases has attracted increasing research interest [1, 29]. The key is to factorize the clicks into relevance probabilities and observation probabilities and model a user's observation probability on an item in a ranking list. By reweighing the click signals based on the reciprocal of observation probabilities, ULTR provides an unbiased estimate of the ranking objective.

Most of the existing ULTR methods are based on the position bias hypothesis [29, 4, 43, 17, 3, 14], which assumes that the observation only depends on the position. We refer to this method as *group-level* ULTR method, in which various heterogeneous features (typically encoded with the

---

*Equal contribution.
†Corresponding authors.

36th Conference on Neural Information Processing Systems (NeurIPS 2022).

information of query, document, and user) share the same observation probability according to some property. However, the observation bias inherently varies across individuals. For example, the exact match in result titles and abstracts (which is usually shown in a different presentation style) affects users' attention and judgment, known as attractive bias [45, 7]. Except for attractive bias, there are lots of factors that can influence a user's observation [10, 47, 37, 7, 24, 31, 32]. These factors are encoded in (or can be inferred from) the features, which leads to heterogeneous observation.

To be in line with these real phenomena, we argue there is an urgent need to develop an *individual-level* ULTR, which can capture the effect of heterogeneous features for observation behavior. That is to say, the observation model should be a function related to documents.

Unfortunately, this problem is very challenging. As shown in Figure 1, there exists a *coupling effect* from features to observation and relevance. There are infinite solutions to factorize the click rate into the relevance function and observation function (all of them are functions of features), but few of them imply a correct ranking model. In the extreme case, a naive ranking model which outputs identical relevance

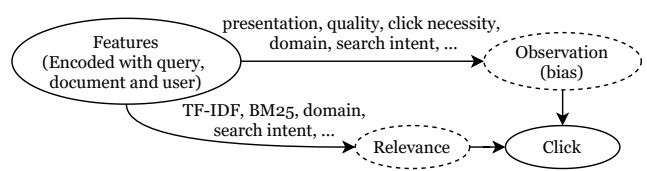

Figure 1: The graphical model when individual-level observation bias exists.

probability still has a chance to perfectly fit the click data with a properly designed association between features and observation [41]. A straightforward solution to avoid coupling is to divide the features manually into two separate parts: one related to relevance and the other related to observation. But it's intractable in practice because some of these features affect both. For example, the exact match in result titles and abstracts affects observation (attractive bias), but on the other hand, it is naturally an important feature encoding how relevant the document is. Besides, result domain [7] and users' search intent [37] are also common factors affecting both.

To address the coupling effect problem, we find that it is unnecessary to correctly estimate the true relevance probability (named hard decoupling), yet only need to recover the correct relevance order from click data (named soft decoupling) because of the ranking objective. Given that we don't know the true relevance, this goal is difficult to achieve directly on the ranking model. We instead modify the observation model, which helps to achieve soft decoupling when the click probability estimation is unbiased. We propose two special observation models, the Lipschitz Observation Model (LOM), which has a Lipschitz constraint $\beta$ on the observation function, and the Bernoulli Observation Model (BOM) which randomly cancels debiasing operation with a probability of $t$ on the observation function output. The two models act on different mechanisms, thus they are complementary and can be combined. We provide a theoretical guarantee that when $\beta$ and $t$ are bounded, these two observation models can achieve soft decoupling whenever estimating unbiased click probability with a ranking model. Given this, we propose a principled novel individual-based ULTR framework, called Lipschitz and Bernoulli Decoupling (LBD) model. We conducted comprehensive experiments on two LTR benchmark datasets, which shows that the proposed LBD outperforms the baseline methods and verifies its effectiveness in debiasing data. We further studied the different application areas of the two decoupling techniques and found that the performance can be better when they are combined.

To the best of our knowledge, we are the first to study the coupling effect for ULTR and propose a feasible solution to decouple the effects. The main contributions of this work are:

1. We present the concept of coupling effect for individual-level ULTR, as well as two targets to achieve decoupling.

2. We propose a novel Lipschitz and Bernoulli Decoupling model, which could help to decouple the effects on relevance and observation. We provide theoretical proof to guarantee the decoupling ability of our proposed techniques.

3. We conducted comprehensive experiments on two LTR benchmark datasets in the coupling effect scene, which shows that the two decoupling techniques have different application areas, and combining them can outperform the baseline methods.

## 2 Related Work

**Debiasing Click Data for LTR**. Debiasing click data is an important research direction in the information retrieval field. We review this line of related work in § A.

**Individual-Level Observation**. There is much work finding that the observation bias can vary from instance to instance. The user's observation can be influenced by result type [32], result quality [10], search intent [37], result domain [7, 24], exact match [45], and even the relevance itself [31]. These factors can be inferred from document features. Particularly, [44] found that in mobile search, the user may be directly satisfied with the result without clicking any links on certain kinds of documents. [37] argued that users with different search intent might have different examination behavior. All these discoveries suggest the necessity to estimate the bias at the individual level.

Recently, researchers attempted to find solutions in ULTR fields to deal with individual-level observation bias. [23] proposed to manually impose a stronger regularization on the position bias estimated by Unbiased LambdaMART, to alleviate the influence of user's different click behaviors, while it's heuristic and has no theoretical proof. [30] proposed wLambdaMART that estimates the confidence of click data with a few labeled data to correct bias, without the need to design a specific bias type. But this method requires extra labeled data. [39] employed heterogeneous treatment effect estimation techniques to estimate the bias between the position $k$ and the position 1. However, this method is based on the assumption that the click rates at the first position reveal their accurate relevance. This is only true when the observation probability at the first position is constant.

## 3 Preliminaries

In this work, we use bold-faced letters to denote vectors, upper-case letters to denote random variables, and the corresponding lower-case letters to denote values. $P(\cdot)$ denotes the distribution of a random variable.

Generally, the core of LTR is to learn a ranking model $f$, which assigns a relevance score to a document with its query-document features $\boldsymbol{X} \in \mathbb{R}^l$. Then, the documents for a query can be sorted in descending order of their scores. In traditional LTR, the ranking model is directly learned with the true relevance score $R$ to make the ranking order of the query list close to the optimal order [29]. In the click setting, users' click logs are used as substitutes of relevance due to their abundance and cheapness. While click signals are naturally biased from the true relevance [27], much prior work follows the **examination hypothesis** to address it, which can be formulated as:

$$c_p(\boldsymbol{x}) = r(\boldsymbol{x}) \cdot o_p(\boldsymbol{x}), \tag{1}$$

where $r(\boldsymbol{x}) = \Pr(R = 1 \mid \boldsymbol{X} = \boldsymbol{x})$, $o_p(\boldsymbol{x}) = \Pr(O = 1 \mid \boldsymbol{X} = \boldsymbol{x}, P = p)$ and $c_p(\boldsymbol{x}) = \Pr(C = 1 \mid \boldsymbol{X} = \boldsymbol{x}, P = p)$ denote the probabilities of relevance, observation and click, and $P$ denotes the position. By explicitly modeling the bias effect via observation probability, it is able to attain an unbiased estimate of the ranking objective.

## 4 Our method

In this section, we first introduce the coupling effect in § 4.1, and propose two decoupling methods in § 4.2 and § 4.3.

### 4.1 Coupling Effect

Let $\hat{r}(\boldsymbol{x}; \theta_1)$ denote the estimation of ranking model, and $\hat{o}(\boldsymbol{x}; \theta_2)$ denote the estimation of observation model. We use clicks to train the two models using Eq.(1), to let their production close to the click. What we care about is if the relevance model is correct when the model converges. To simplify the later analysis, we suppose that the model has a universal approximation capacity [22] to predict an unbiased click probability, and make the following assumption.

**Assumption 1** (Unbiased Click Prediction). *There exists a parameter space $\Theta^*$ which consists of pairs of ranking model parameter and observation model parameter, such that for all $(\theta_1^*, \theta_2^*) \in \Theta^*$, the predictive click probability is unbiased:*

$$\hat{r}(\boldsymbol{x}; \theta_1^*) \cdot \hat{o}_p(\boldsymbol{x}; \theta_2^*) = c_p(\boldsymbol{x}), \quad \forall \boldsymbol{x} \in \mathbb{R}^l, p \in [n].$$

From Assumption 1 we can find that the factorization of relevance probability $\hat{r}(\boldsymbol{x}; \theta_1^*)$ and observation probability $\hat{o}_p(\boldsymbol{x}; \theta_2^*)$ is unidentifiable given the click probability $c_p(\boldsymbol{x})$, which makes it challenging to uncover the correct ranking model. This is because the features $\boldsymbol{X}$ have an effect on both relevance $R$ and observation $O$. It's difficult to distinguish which parts of the features affect relevance and observation, respectively. In the extreme case, the ranking model will degenerate into a trivial case that always predicts the same scores for all documents [41]. We refer to this problem as the *coupling effect*.

Suppose the correct factorization is $c_p(\boldsymbol{x}) = r(\boldsymbol{x}) \cdot o_p(\boldsymbol{x})$, where $r(\boldsymbol{x})$ is the true relevance and $o_p(\boldsymbol{x})$ is the true observation. To achieve a good decoupling of the features' effect on relevance and observation, we aim to make $\hat{r}(\boldsymbol{x}; \theta_1)$ close to $r(\boldsymbol{x})$ whenever predicting an accurate click probability. Based on this, we introduce the definition of **hard decoupling** as follows.

**Definition 1** (Hard Decoupling). *For any features $\boldsymbol{x}$ with the true relevance $r(\boldsymbol{x})$, a parameter space $\Theta' \subseteq \Theta^*$ can do hard decoupling, if for all $(\theta_1^*, \theta_2^*) \in \Theta'$, we have $r(\boldsymbol{x}) = \hat{r}(\boldsymbol{x}; \theta_1^*)$. $\Theta^*$ is defined in Assumption 1.*

Considering the pairwise comparison of the ranking objective, we further relax the hard decoupling constraint and refer to this as **soft decoupling** as follows.

**Definition 2** (Soft Decoupling). *For any two features $\boldsymbol{x}_1$ and $\boldsymbol{x}_2$, without loss of generality we suppose $r(\boldsymbol{x_1}) \geq r(\boldsymbol{x_2})$. A parameter space $\Theta' \subseteq \Theta^*$ can do soft decoupling, if for all $(\theta_1^*, \theta_2^*) \in \Theta'$, we have $\hat{r}(\boldsymbol{x_1}; \theta_1^*) \geq \hat{r}(\boldsymbol{x_2}; \theta_1^*)$. $\Theta^*$ is defined in Assumption 1.*

Soft decoupling requires identifying the relevance, which is challenging when relevance is unobserved. In this subsection, we propose two mechanisms (Lipschitz Decoupling and Bernoulli Decoupling) for soft decoupling with theoretical guarantees. From this point forward, we assume that all of the parameters $\theta$ take the value from $\Theta^*$, and omit $\theta$ of $\hat{r}(\cdot; \theta)$ and $\hat{o}_p(\cdot; \theta)$ for ease of notation.

## 4.2 Lipschitz Decoupling

Usually, the rate of change between the observation and features will not be substantial in real scenarios. Therefore, we propose modifying the observation model's smoothness to make it match the true observation function, which helps to achieve soft decoupling.

We start with defining the Lipschitz continuity for a function. Let $||\cdot||$ be a norm on $\mathbb{R}^l$. In this work, we employ the Euclidean norm (L-2) when not mentioned otherwise: $||\boldsymbol{x}|| = ||\boldsymbol{x}||_2 = \sqrt{\sum_{i=1}^l x_i^2}$.

**Definition 3** (Lipschitz Continuity). *A function $f(\boldsymbol{x}) : \mathbb{R}^l \to \mathbb{R}$ is $\alpha$-Lipschitz to the input $\boldsymbol{x}$ if*

$$|f(\boldsymbol{x}') - f(\boldsymbol{x})| \leq \alpha ||\boldsymbol{x}' - \boldsymbol{x}||,$$

*where $\alpha$ is referred to as the Lipschitz constant.*

Particularly, if an $\alpha$-Lipschitz function $f$ is differentiable, we have $||\nabla_{\boldsymbol{x}} f(\boldsymbol{x})|| \leq \alpha$. We suppose $r(\boldsymbol{x}), o_p(\boldsymbol{x}), \hat{r}(\boldsymbol{x})$ and $\hat{o}_p(\boldsymbol{x})$ are all differentiable.

**Assumption 2** (True Observation Lipschitz). *For each $p \in [n]$, the true observation $o_p(\boldsymbol{x})$ is $\alpha$-Lipschitz.*

It assumes that there is an upper bound on the rate at which the true observation changes with the features. This is a reasonable extension for well-known position-based model [29], which assumes that $\alpha = 0$. Similarly, we assume the observation estimation is also Lipschitz. We refer to this kind of observation model as **Lipschitz Observation Model** (LOM).

**Definition 4** (Lipschitz Observation Model). *An observation model is $\beta$−Lipschitz observation model ($\beta$-LOM), if its estimation $\hat{o}_p(\boldsymbol{x})$ is $\beta$-Lipschitz for all $p \in [n]$.*

We claim that soft decoupling can be achieved by selecting a lower $\beta$, as the following theorem.

**Theorem 1** (Lipschitz Decoupling). *For any two features $\boldsymbol{x}_1$ and $\boldsymbol{x}_2$, without loss of generality we suppose $r(\boldsymbol{x_1}) \geq r(\boldsymbol{x_2})$. For a $\beta$-LOM, if the following condition is satisfied,*

$$\frac{r(\boldsymbol{x_1})}{r(\boldsymbol{x_2})} - 1 \geq \frac{o_p(\boldsymbol{x_1})}{\hat{o}_p(\boldsymbol{x_1})} \left\{ \inf_{\gamma \in \Gamma(\boldsymbol{x}_1, \boldsymbol{x}_2)} \int_\gamma \frac{\beta o_p(\boldsymbol{x}) + \alpha \hat{o}_p(\boldsymbol{x})}{o_p^2(\boldsymbol{x})} ||d\boldsymbol{x}|| \right\},$$

where $\Gamma(\boldsymbol{x}_1, \boldsymbol{x}_2)$ denotes the collection of all curves from point $\boldsymbol{x}_1$ to point $\boldsymbol{x}_2$. Then we have: $\hat{r}(\boldsymbol{x_1}) \geq \hat{r}(\boldsymbol{x_2})$.

**Remark 1.** *If the gap between $r(\boldsymbol{x_1})$ and $r(\boldsymbol{x_2})$ is large (i.e., $\frac{r(\boldsymbol{x_1})}{r(\boldsymbol{x_2})}$ is large) and $\alpha$ is small enough, there exists a $\beta$ such that the condition can be satisfied. Namely, when the true observation doesn't change too fast with the features, we can select a lower $\beta$ to achieve soft decoupling.*

At a high level, $\beta$-LOM controls the capacity of the function space $\Theta^*$ by restricting the Lipschitz constant $\beta$. While a low $\beta$ benefits the decoupling, it also suffers from the risk of underfitting. For example, if we adopt a 0-LOM (known as the Position-based Model) in the setting that the true observation has a large Lipschitz constant, we may not find a $\hat{r}(\boldsymbol{x})$ and a $\hat{o}_p(\boldsymbol{x})$ such that their product is equal to a given click rate $c_p(\boldsymbol{x})$. In other words, $\beta$ has a lower bound. This can be shown formally in the following theorem.

**Theorem 2** (Lower Bound of $\beta$). *Let $\boldsymbol{c}(\boldsymbol{x})$ denote a vector containing click rates at each position:*

$$\boldsymbol{c}(\boldsymbol{x}) = (r(\boldsymbol{x})o_1(\boldsymbol{x}), r(\boldsymbol{x})o_2(\boldsymbol{x}), \ldots, r(\boldsymbol{x})o_n(\boldsymbol{x}))^\top .$$

*For any two different features $\boldsymbol{x}_1$ and $\boldsymbol{x}_2$, we have*

$$\beta \geq \inf_{m,n \geq 1} \frac{||m\boldsymbol{c}(\boldsymbol{x}_1) - n\boldsymbol{c}(\boldsymbol{x}_2)||_\infty}{||\boldsymbol{x}_1 - \boldsymbol{x}_2||_2}.$$

**Remark 2.** *We consider a special situation where the true observation is 0-Lipschitz ($\alpha = 0$). In this case, o is no longer a function of $\boldsymbol{x}$ but a constant, thus we can find $m, n$ such that $m\boldsymbol{c}(\boldsymbol{x}_1) - n\boldsymbol{c}(\boldsymbol{x}_2)$ is a zero vector, and the lower bound of $\beta$ can be zero. When the true observation has a large Lipschitz number, $m\boldsymbol{c}(\boldsymbol{x}_1) - n\boldsymbol{c}(\boldsymbol{x}_2)$ can no longer be a zero vector and $\beta$ should be larger. It means the models should change quickly with the input feature to fit clicks.*

Theorem 1 and Theorem 2 give the upper bound and lower bound of $\beta$, respectively. The proofs of these two theorems are provided in Appendix § B. Since the bounds focus on the pairwise comparison of documents and it's difficult to find the best parameter for the global dataset, in practice we can tune the value of $\beta$ to achieve the best global effect for soft decoupling. We refer to this method as **Lipschitz Decoupling**.

### 4.3 Bernoulli Decoupling

Unfortunately, when $\alpha$ is large in a given dataset, it's difficult to select a proper $\beta$ such that the upper bound and the lower bound are satisfied at the same time. Under this circumstance, it may perform even worse than that without any debiasing operation (see § 6.3 for discussion). As a complementary, we propose another decoupling technique, named Bernoulli Decoupling. Unlike Lipschitz Decoupling, which makes constraints on the observation model's estimation $\hat{o}_p(\boldsymbol{x}; \theta_2)$, we explicitly add an additional sampling step to transform its output, as follows:

$$\gamma \sim \text{Bernoulli}(1 - t), \quad \hat{o}_p'(\boldsymbol{x}; \theta_2, t) = 1 + (\hat{o}_p(\boldsymbol{x}; \theta_2) - 1)\gamma.$$

That is to say, we randomly cancel the debiasing operation with a certain probability $t$ (i.e., force the observation model to predict 1 as an observation score and let the ranking model learn directly from clicks). Since the features are generally collected for the ranking task, the effects from features to relevance are usually stronger than the ones from features to observation. Thus, we hope the relevance model could learn more effects from features to clicks than the observation model. The Bernoulli sampling prevents the relevance model from over-relying on a wrong observation model and encourages the relevance model to relearn what was learned by the observation model. The probability $t$ can be seen as a global parameter to control the strength of relevance effects. Similar to our method, Zhao et al.[46] also uses a dropout technique to prevent the model over-relying on the position. We define the observation model based on this new sampling process as $t$-**Bernoulli Observation Model** ($t$-BOM).

For convenience, we denote the expectation $\mathbb{E}(\hat{o}_p'(\boldsymbol{x}; \theta_2, t))$ as the estimation of the observation model after transformation, and Assumption 1 holds based on this expectation. We can prove that by choosing a sufficiently large $t$, BOM can achieve soft decoupling:

**Theorem 3** (Bernoulli Decoupling)**.** *For any two different features $\boldsymbol{x_1}$ and $\boldsymbol{x_2}$, without loss of generality we suppose $r(\boldsymbol{x_2}) \geq r(\boldsymbol{x_1})$. For a $t-$BOM, if the following condition is satisfied,*

$$t \geq \frac{r(\boldsymbol{x_1})}{r(\boldsymbol{x_2})} \left( 1 + \frac{\alpha ||\boldsymbol{x_1} - \boldsymbol{x_2}||}{\sup_p o_p(\boldsymbol{x_2})} \right),$$

*then we have: $\hat{r}(\boldsymbol{x_2}) \geq \hat{r}(\boldsymbol{x_1})$.*

**Remark 3.** *If the true observation $o_p(\boldsymbol{x})$ is 0-Lipschitz (i.e., $\alpha = 0$), there must exist a $t \leq 1$ making the above error condition satisfied to achieve soft decoupling. For a larger Lipschitz number, we can increase the value of $t$ appropriately.*

Similar to LOM, BOM is another method to control the capacity of $\Theta^*$, and it also suffers from the risk of underfitting. In order to make it fit the click rates, $t$ has an upper bound.

**Theorem 4** (Upper Bound of $t$)**.** *For any feature $\boldsymbol{x}$ and a $t-$BOM, we have*

$$t \leq \frac{\inf_p o_p(\boldsymbol{x})}{\sup_p o_p(\boldsymbol{x})}.$$

Theorem 3 and Theorem 4 give the lower bound and the upper bound of $t$ respectively. The proofs of the theorems are provided in Appendix § B. Like LOM, we take $t$ as a tuning parameter in practice. We refer to this method as **Bernoulli Decoupling**.

Bernoulli Decoupling is a complement to Lipschitz Decoupling, and they can be used simultaneously to improve the efficiency of decoupling. This is because the range in which the two hyperparameters take effect is an "or" relationship: Even if $\beta$ is outside the scope of Theorem 1 and Theorem 2, as long as $t$ is within the scope of Theorem 3 and Theorem 4, the combined model can still achieve soft decoupling. This shows that it is beneficial to combine the two models together.

Finally, we propose our $(\beta, t)$**-Lipschitz and Bernoulli Decoupling** (LBD) model, which combines the $\beta$-LOM and $t$-BOM. We treat LBD as our complete solution for individual-based ULTR.

## 5 Model Implementation

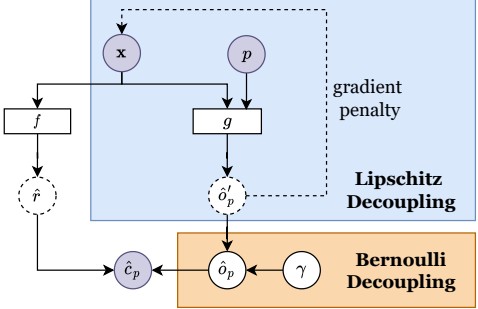

Figure 2: Framework of the proposed LBD.

Up to this point, we have shown that a $(\beta, t)$-LBD can effectively decouple clicks on a relatively low-Lipschitz observation dataset. In this section, we first introduce the implementation of the proposed LBD, then present the optimization objective of the proposed model. The overall model architecture is illustrated in Figure 2.

We first use a neural network $f$ as the ranking model (take features as input) and a neural network $g$ as the observation model (take features and position as inputs), and output the relevance and the observation: $\hat{r} = f(\boldsymbol{x}), \hat{o}_p = g_p(\boldsymbol{x})$.

**Lipschitz Decoupling**  Based on Theorem 1 and 2, we need to enforce a proper Lipschitz constraint on the observation model to employ Lipschitz Decoupling. Since we use a neural network to implement the observation model, it's difficult to directly control the Lipschitz constant without adjusting the network structure too much. As an easy alternative, [19] proposed to add a penalty on the gradient norm. Similar to them, we employ a gradient penalty loss function: $L_{gp}(\boldsymbol{x}) = \lambda \sum_{i=1}^{n} ||\nabla_{\boldsymbol{x}}(\hat{o}_i)||_2$. This is based on the fact that a differentiable function is $k$-Lipschtiz if and only if it has gradients with the norm at most $k$ everywhere. The gradient penalty loss encourages the model to have a low Lipschitz constant, and we can adjust $\lambda$ to control the penalty level and obtain models with different Lipschitz constants. Thus, $\lambda$ can be a substitute for $\beta$.

**Bernoulli Decoupling**  Based on Theorems 3 and 4, in order to implement the Bernoulli Decoupling technique and avoid degradation of the debiasing process, we sample a $\gamma$ from Bernoulli distribution parameterized by $1-t$. Our final logarithmic observation probability is generated by $\log \hat{o}'_p = \gamma \log \hat{o}_p$, where $\gamma \sim \text{Bernoulli}(1-t)$. The final predictive click score is: $\log \hat{c}_p = \log \hat{r} + \log \hat{o}'_p$.

**Objective Function and Training Process**  Similar to [4], we adopt a list-wise loss based on softmax cross entropy. Given a query $q$ with the document ranking list $\pi_q = \{\boldsymbol{x}_1, \boldsymbol{x}_2, \cdots, \boldsymbol{x}_n\}$, we generate click $\hat{c}_p$ for feature $\boldsymbol{x}_p$ at each position $p$. Suppose the real click signals for each document in $\pi_q$ are $c_1, c_2, \cdots, c_n$, we use the following supervise loss to train our models:

$$L_{sv}(q) = -\sum_{i=1}^{n} c_i \frac{\exp(\log \hat{c}_i)}{\sum_{j=1}^{n} \exp(\log \hat{c}_j)}. \tag{2}$$

Finally, the objective function based on a query $q$ can be written as $L(q) = L_{sv}(q) + \sum_{i=1}^{n} L_{gp}(\boldsymbol{x}_i)$. We jointly update the ranking model and the observation model with the derivatives of $L(q)$ and repeat the process until the algorithm converges.

# 6  Experiments

In this section, we describe our experimental setup and show the empirical results.

## 6.1  Experimental Setup

**Dataset**  We followed the standard setup in CLTR [5, 4, 15, 39] and conducted semi-synthetic experiments on two widely used benchmark datasets: Yahoo! LETOR[3] [12] and Istella-S[4] [33]. We provide further details for these datasets in Appendix § C.1. We followed the given data split of training, validation and testing. To generate initial ranking lists for click simulation, we followed the standard process [29, 4, 15] to train a Ranking SVM model [26] with 1% of the training data with relevance labels, and sort the documents. Based on these initial ranking lists, we sampled clicks according to the examination hypothesis. Following the steps proposed by [13], we set the relevance probability to be:

$$\Pr(R = 1 \mid \boldsymbol{X} = \boldsymbol{x}) = \epsilon + (1 - \epsilon) \frac{2^{y_{\boldsymbol{x}}} - 1}{2^{y_{\max}} - 1}, \tag{3}$$

where $y_{\boldsymbol{x}} \in [0, y_{\max}]$ is the relevance level of $\boldsymbol{x}$ by human annotation, and $y_{\max} = 4$ in both of the two datasets. $\epsilon$ is the click noise level and we set $\epsilon = 0.1$ as the default setting.

To the best of our knowledge, we are the first to study the coupling effects, thus there is no precedent for the observation simulation. In order to simulate the observation, We first select some crux features as the factors that are related to observation, such as result type, result quality or search intent, etc. Because the exact meaning of the features given by the datasets is unknown, we follow the method in previous work [39, 15] to select some features as substitutes: (1) use normalized features and relevances in the total dataset to train ExtRa Trees [18]; (2) based on this model, select and combine the top-10 important features as crux features. We adopted the methodology in [39] to model the feature's influence on observation. The observation probability is set to be:

$$\Pr(O = 1 \mid \boldsymbol{X} = \boldsymbol{x}, P = p) = v_p^{\max\{\boldsymbol{w}^\top f_{\text{sel}}(\boldsymbol{x}') + 1, 0\}},$$

---

[3]https://webscope.sandbox.yahoo.com/
[4]http://quickrank.isti.cnr.it/istella-dataset/

where $v_p$ is the position-based examination probability at position $p$ by eye-tracking studies [27]. $\boldsymbol{w}$ is a 10-dimensional vector uniformly drawn from $[-\eta, \eta]$, where $\eta$ is a hyperparameter to control the dependency between the observation and crux features. $\boldsymbol{x}'$ denotes the crux features of $\boldsymbol{x}$. Note that $\eta$ can be explained as the coupling level between relevance and observation.

**Baselines**   We combined the state-of-the-art ULTR methods and two LTR models for comparison. Debiasing methods include **Vectorization** [14], **RegressionEM** [43], **DLA** [4], **PairDebias** [23], **HTE** [39], **Labeled Data** (uses human-annotated relevance labels to train the ranker directly) and **Click Data** (uses the raw click data to train the ranker directly). Ranking models include **DNN** and **Linear**. We adopted the codes in ULTRA framework [5, 6] to implement RegressionEM, DLA, PairDebias and the ranking models, and kept the same hyperparameters. Note that Regression-EM, DLA and PairDebias are all group-level ULTR methods. HTE is an individual-level ULTR, but it's not a jointly learning algorithm, so we compared it in another scene.

Our method is referred to as **LBD**. We also use three degenerate versions for ablation study: (1) set $t = 0$ to remove the Bernoulli sampling step (named as **LBD**$_{Lips}$), (2) set $\lambda = 0$ to remove the gradient penalty loss function (named as **LBD**$_{Ber}$), and (3) remove both (named as **Unlimited**). Training details can be found in § C.2.

### 6.2   Performance on different rankers and datasets

Table 1: nDCG@10 performance of different methods on two datasets ($\eta = 0.1$). Significant performance (p-value $< 0.005$) improvement/degradation compared to the best baseline (except for Labeled Data) by t-test is denoted as +/-. Table 4 in the appendix shows the full results.

| Dataset | Yahoo! | | Istella-S | |
|---|---|---|---|---|
| Ranker | DNN | Linear | DNN | Linear |
| Labeled Data | 0.764 | 0.747 | 0.729 | 0.689 |
| DLA | 0.755 | 0.741 | 0.696 | 0.667 |
| Vectorization | 0.751 | 0.741 | 0.691 | 0.672 |
| PairDebias | 0.739 | 0.738 | 0.691 | 0.678 |
| RegressionEM | 0.746 | 0.720 | 0.623 | 0.594 |
| Click Data | 0.741 | 0.737 | 0.701 | 0.680 |
| Unlimited | 0.752 | $0.719^-$ | $0.687^-$ | 0.592 |
| LBD$_{Lips}$ | $0.757^+$ | $0.744^+$ | $0.686^-$ | $0.659^-$ |
| LBD$_{Ber}$ | 0.754 | 0.742 | $0.707^+$ | 0.679 |
| LBD | $\mathbf{0.758}^+$ | $\mathbf{0.746}^+$ | $\mathbf{0.709}^+$ | **0.680** |

We first study the performance of jointly learning-based ULTR models. Table 1 summarizes the NDCG@10 results about the performance of different rankers on different datasets. We see that our proposed LBD significantly outperforms almost all the other baseline methods since we properly handle the coupling effect. The results of other baselines are compatible with those reported in [6]. Particularly, we have the following findings:

1. LBD works better than LBD$_{Lips}$ and LBD$_{Ber}$, which indicates that using Lipschitz Decoupling and Bernoulli Decoupling at the same time is better than using them alone. The reason seems to be that the two decoupling methods have different conditions (see Theorem 1 and Theorem 3). When using them together, the scope of decoupling can be expanded.

2. Unlimited works worst than almost all the best baselines, indicating the existence of the coupling effect. If we train the models unlimitedly, the performance would be quite bad. Additionally, the coupling effect is more serious when using the Linear ranker.

3. Lipschitz Decoupling performs better than Bernoulli Decoupling on Yahoo! dataset, while the opposite is true on the Istella-S dataset. It shows that the application of two decoupling methods is tightly related to the data distribution.

### 6.3   Performance on different coupling levels

In this experiment, we investigate the performance under different degrees of coupling level on the Yahoo! dataset, where the ranker is DNN. We vary $\eta$ from 0 to 0.6 when generating clicks.

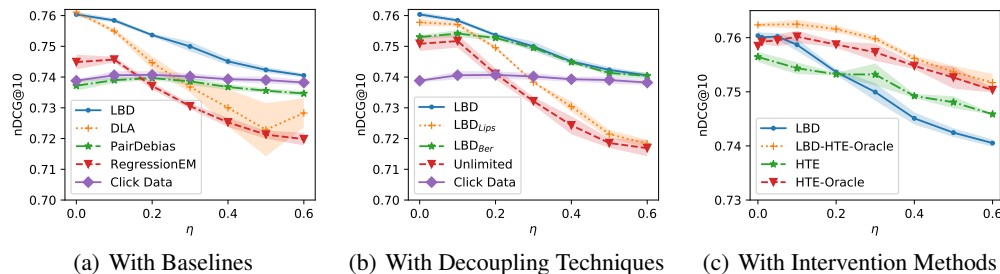

| (a) With Baselines | (b) With Decoupling Techniques | (c) With Intervention Methods |

Figure 3: Performance across different coupling levels $\eta$. The larger the value of $\eta$, the stronger the influence of features on observation. The variance is displayed with the shadow areas.

Figure 3(a) shows the influence of coupling levels on different offline ULTR methods. We can observe that our model achieves the best performance under all degrees of coupling level. When the observation is 0-Lipschitz (i.e., $\eta = 0$), our model and DLA perform similarly, since both of them employ a similar listwise loss function. The debiasing performance of other baselines diminishes with the increased coupling level. For a large coupling level ($\eta \geq 0.3$), all baselines perform worse than the raw click data. In contrast, our proposed method always performs better than click data, which shows the strong decoupling ability of LBD. Besides, our method exhibits a downward trend with the increased coupling level, which demonstrates that it's more difficult to decouple the effects when the true observation has a large Lipschitz constant.

Figure 3(b) shows the performance of different decoupling methods we propose in this work. One can see that for a low degree of coupling level ($\eta < 0.2$), $LBD_{Lips}$ performs better than $LBD_{Ber}$, which implies that LOM can decouple the effects under a low Lipschitz observation dataset. For a large coupling level, LOM is worse than BOM in turn. One explanation is that BOM can avoid the deterioration of ULTR when the relevance and observation effects are difficult to decouple since it randomly drops out the debiasing operation. Besides, LBD always performs better than $LBD_{Lips}$ and $LBD_{Ber}$, once again showing that using them together could expand the scope of decoupling.

The previous study focuses most on joint-learning ULTR methods. To compare the Heterogeneous Treatment Effect (HTE), we conduct another experiment based on intervention data. Following [39], we simulate 2,560,000 queries for the intervention experiment before training the ranking model (this number is equal to the number of total queries we used to train models). For each query, we randomly chose a $k \in [n]$ and swapped the document at the first position with the one at the $k$-th position. We collected clicks generated on these ranking lists and calculated the HTE estimator $\hat{\tau}_k(\boldsymbol{x})$. Based on $\hat{\tau}_k(\boldsymbol{x})$, we debiased the click data and use it to train a ranking model directly. We also used an accurate HTE estimator $\tau_k(\boldsymbol{x})$ from the click model we used, which can be regarded as the upper bound of the HTE method. Based on the clicks corrected with $\tau_k(\boldsymbol{x})$, we train a ranking model directly (named HTE-oracle). We also use these clicks to train our model (named LBD-HTE-oracle).

We present the results in Figure 3(c). We can observe that our LBD (trained without any intervention data) performs even better than HTE and HTE-oracle in a low degree of coupling level because HTE is based on an assumption that the click rates at the first position can reveal the correct relevance, which is biased when the observation probability is not constant. For a high degree of coupling level, HTE performs better than our method. One possible reason is that intervention can augment the dataset (since each document has an opportunity to display at the top position), which is helpful when the effects are difficult to decouple. Besides, when our model is trained on the clicks corrected with the oracle HTE, it can always outperform the vanilla HTE method. This demonstrates once again that HTE is still biased and verifies our decoupling effectiveness.

## 6.4 Hyper Parameter Analysis

From the previous discussion, we know that the hyperparameters should have an upper bound and lower bound depending on the data. To investigate it, we study the performance with different hyperparameters. Figure 4 shows the results when changing hyper parameters. We can see that there is a focal point on each heat map, and the grid points around the focal point have a lower performance,

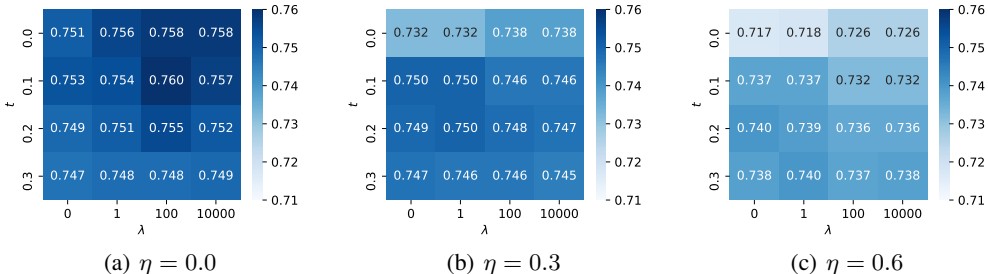

Figure 4: Performance with different hyper parameters across different coupling levels. A darker color indicates a better nDCG@10 performance.

which verifies our discussion. Besides, the focal point gradually shifts from the upper right corner to the lower-left corner as an increase. This shows again that Lipschitz Decoupling takes a better effect on a low Lipschitz observation dataset, while Bernoulli Decoupling performs better on a large Lipschitz observation dataset.

## 7    Conclusion

In this work, we take the first step to studying the coupling effect in individual-level ULTR. We propose two goals: hard decoupling, where the true relevance probability should be estimated accurately, and soft decoupling, where the correct relevance order can be recovered. We modify the observation model to achieve soft decoupling when the click probability estimation is unbiased. We propose two special observation models, Lipschitz Observation Model (LOM) and Bernoulli Observation Model (BOM). We provide a theoretical guarantee that when their hyperparameters are bounded, soft decoupling can be achieved by these methods. We conduct comprehensive experiments on two LTR benchmark datasets, which shows that the proposed LBD outperforms the baseline methods and verifies its effectiveness in debiasing data.

**Limitations.** (1) For simplicity, our theoretical results are based on Assumption 1, which requires the models to fit clicks perfectly. It would be interesting to investigate how the decoupling effect will be affected if the click estimation is not so accurate. (2) Our bounds in the theorems contain a value $\alpha$ from Assumption 2 which is related to the real data generating process. In some cases such as sparse, high dimensional, or high noise cases, the value of alpha may be too large and invalidate the derived bounds. How to decouple data in these cases is an open question.

## Societal Impact

To the best of our knowledge, the approaches in this paper raise no major ethical concerns or societal consequences. Researchers and practitioners from the ULTR domain may benefit from our research since debiasing implicit feedback is a significant challenge in real-world applications. The worst possible outcome when the proposed approach fails is that it reduces to the standard position-based observation estimation and stops making the desired impact. Finally, the proposed approach aims at solving the coupling effects of the data, the extent of which depends on the properties of the data.

## Acknowledgments

Thanks to Yusu Hong for the discussion on the theorems, and the reviewers for their valuable comments and suggestions.

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
