# Appendix

## A   Extended Related Work

There are two groups of approaches to debias click data for ranking. The first is to model user's behavior to infer relevance from biased click signals, known as *click models* [16, 21, 20, 9, 8]. However, most click models focus on predicting clicks, rather than optimizing the ranking performance [29, 4]. They are usually separated from the LTR frameworks, and the relevance inference is an afterthought [4]. The second group tries to directly learn unbiased ranking models from biased clicks, known as *unbiased learning to rank* (ULTR). Joachims et al. [29] proposed the inverse propensity scoring (IPS) method to reweigh the click signals based on the reciprocal of observation probabilities (called propensity scores) and provide an unbiased estimate of the ranking objective. The propensity scores are estimated by randomized experiments [29, 42], which hurt users' experience, unfortunately. To address it, Agarwal et al. [3] and Fang et al. [17] proposed to do intervention harvest by exploiting click logs with multiple ranking models. Nevertheless, they have a relatively narrow scope of application due to the strict assumption to construct interventional sets. Recently, some researchers proposed to jointly estimate relevance and bias [43, 4, 23, 25, 14]. Similar to them, our proposed method could jointly train the ranking model and observation model without intervention.

On the other side, researchers developed models to deal with all kinds of click bias, based on ULTR framework. According to types of bias, they can be divided into several categories: (1) position bias, where the bias only depends on position [43, 4, 23, 11, 14]; (2) selection bias, where some documents have a zero probability of being observed since it is ranked below a certain cutoff [34, 35, 36]; (3) trust bias, where users are more likely to click incorrectly on higher-ranked items [2, 41]; (4) contextual bias, where the observation bias varies from query to query [17, 39]; (5) interactional bias, where the observation is influenced by interactions among clicks in the same ranking list [15, 40]. However, these types of work mainly model the observation bias at a group level and ignore the characteristics of each document.

## B   Proofs of the Theorems

### B.1   Proof of Theorem 1

Let $\sigma_p(\boldsymbol{x}) = \frac{\hat{o}_p(\boldsymbol{x})}{o_p(\boldsymbol{x})}$, for each $p \in [n]$ we have:

$$
\begin{aligned}
\beta &\geq ||\nabla_{\boldsymbol{x}}\left(\hat{o}_p(\boldsymbol{x})\right)|| \\
&= ||\nabla_{\boldsymbol{x}}\left(\sigma_p(\boldsymbol{x})o_p(\boldsymbol{x})\right)|| \\
&= ||o_p(\boldsymbol{x})\nabla_{\boldsymbol{x}}\left(\sigma_p(\boldsymbol{x})\right) + \sigma_p(\boldsymbol{x})\nabla_{\boldsymbol{x}}\left(o_p(\boldsymbol{x})\right)|| \\
&\geq ||o_p(\boldsymbol{x})\nabla_{\boldsymbol{x}}\left(\sigma_p(\boldsymbol{x})\right)|| - ||\sigma_p(\boldsymbol{x})\nabla_{\boldsymbol{x}}\left(o_p(\boldsymbol{x})\right)||.
\end{aligned}
$$

This implies the following bounds:

$$
||\nabla_{\boldsymbol{x}}\left(\sigma_p(\boldsymbol{x})\right)|| \leq \frac{\beta + ||\sigma_p(\boldsymbol{x})\nabla_{\boldsymbol{x}}\left(o_p(\boldsymbol{x})\right)||}{o_p(\boldsymbol{x})} \leq \frac{\beta + \alpha\sigma_p(\boldsymbol{x})}{o_p(\boldsymbol{x})}.
$$

Now by using the condition in Theorem 1, we obtain:

$$\frac{\sigma_p(\boldsymbol{x_1})}{\sigma_p(\boldsymbol{x_2})} = 1 + \frac{1}{\sigma_p(\boldsymbol{x_2})}(\sigma_p(\boldsymbol{x_1}) - \sigma_p(\boldsymbol{x_2}))$$

$$= 1 + \frac{1}{\sigma_p(\boldsymbol{x_2})}\int_{\boldsymbol{x_2}}^{\boldsymbol{x_1}} \nabla(\sigma_p(\boldsymbol{x})) \cdot \mathrm{d}\boldsymbol{x}$$

$$\leq 1 + \frac{1}{\sigma_p(\boldsymbol{x_2})}\left\{\inf_{\gamma\in\Gamma(\boldsymbol{x_1},\boldsymbol{x_2})}\int_{\gamma} ||\nabla(\sigma_p(\boldsymbol{x}))||\,||\mathrm{d}\boldsymbol{x}||\right\}$$

$$\leq 1 + \frac{1}{\sigma_p(\boldsymbol{x_2})}\left\{\inf_{\gamma\in\Gamma(\boldsymbol{x_1},\boldsymbol{x_2})}\int_{\gamma} \frac{\beta + \alpha\sigma_p(\boldsymbol{x})}{o_p(\boldsymbol{x})}||\mathrm{d}\boldsymbol{x}||\right\}$$

$$\leq 1 + \frac{1}{\sigma_p(\boldsymbol{x_2})}\frac{\hat{o}_p(\boldsymbol{x_2})}{o_p(\boldsymbol{x_2})}\left(\frac{r(\boldsymbol{x_1})}{r(\boldsymbol{x_2})} - 1\right)$$

$$= 1 + \left(\frac{r(\boldsymbol{x_1})}{r(\boldsymbol{x_2})} - 1\right) = \frac{r(\boldsymbol{x_1})}{r(\boldsymbol{x_2})}.$$

In the above deviation, the second inequality uses the upper bound of $||\nabla_{\boldsymbol{x}}(\sigma_p(\boldsymbol{x}))||$, and the third inequality uses the condition in Theorem 1.

Now by using Assumption 1, we have:

$$\hat{r}(\boldsymbol{x_1}) = \frac{r(\boldsymbol{x_1})o_p(\boldsymbol{x_1})}{\hat{o}_p(\boldsymbol{x_1})} = \frac{r(\boldsymbol{x_1})}{\sigma_p(\boldsymbol{x_1})} \geq \frac{r(\boldsymbol{x_2})}{\sigma_p(\boldsymbol{x_2})} = \frac{r(\boldsymbol{x_2})o_p(\boldsymbol{x_2})}{\hat{o}_p(\boldsymbol{x_2})} = \hat{r}(\boldsymbol{x_2}),$$

where the inequality uses the above conclusion, and the first and the last equality uses Assumption 1. This implies the desired bound.

## B.2 Proof of Theorem 2

Given that this model is a $\beta$-LOM, by using Assumption 1 we have

$$\beta \geq \frac{|\hat{o}_p(\boldsymbol{x_1}) - \hat{o}_p(\boldsymbol{x_2})|}{||\boldsymbol{x_1} - \boldsymbol{x_2}||} = \frac{\left|\frac{c_p(\boldsymbol{x_1})}{\hat{r}(\boldsymbol{x_1})} - \frac{c_p(\boldsymbol{x_2})}{\hat{r}(\boldsymbol{x_2})}\right|}{||\boldsymbol{x_1} - \boldsymbol{x_2}||}, \forall p \in [n],$$

where $c_p(\boldsymbol{x}) = o_p(\boldsymbol{x})r(\boldsymbol{x})$. Now by taking the lower bound on the right side over $p$, we obtain:

$$\exists \hat{r}(\boldsymbol{x_1}), \hat{r}(\boldsymbol{x_2}) \in [0, 1], \text{s.t.,}$$

$$\beta \geq \sup_p \frac{\left|\frac{c_p(\boldsymbol{x_1})}{\hat{r}(\boldsymbol{x_1})} - \frac{c_p(\boldsymbol{x_2})}{\hat{r}(\boldsymbol{x_2})}\right|}{||\boldsymbol{x_1} - \boldsymbol{x_2}||} = \frac{\left\|\frac{\boldsymbol{c}(\boldsymbol{x_1})}{\hat{r}(\boldsymbol{x_1})} - \frac{\boldsymbol{c}(\boldsymbol{x_2})}{\hat{r}(\boldsymbol{x_2})}\right\|_\infty}{||\boldsymbol{x_1} - \boldsymbol{x_2}||}.$$

By taking $m = \frac{1}{\hat{r}(\boldsymbol{x_1})}$ and $n = \frac{1}{\hat{r}(\boldsymbol{x_1})}$, we can obtain the desired result.

## B.3 Proof of Theorem 3

From the condition, we have:

$$t \geq \inf_p \frac{r(\boldsymbol{x_1})}{r(\boldsymbol{x_2})}\left(1 + \frac{\alpha||\boldsymbol{x_1} - \boldsymbol{x_2}||}{o_p(\boldsymbol{x_2})}\right)$$

$$\geq \inf_p \frac{r(\boldsymbol{x_1})}{r(\boldsymbol{x_2})}\left(1 + \frac{|o_p(\boldsymbol{x_2}) - o_p(\boldsymbol{x_1})|}{o_p(\boldsymbol{x_2})}\right)$$

$$\geq \inf_p \frac{r(\boldsymbol{x_1})o_p(\boldsymbol{x_1})}{r(\boldsymbol{x_2})o_p(\boldsymbol{x_2})} \tag{4}$$

Let $\hat{o}'_p(\boldsymbol{x}; t)$ denote the estimation of a $t$-BOM. From the definition, we have:

$$\hat{o}'_p(\boldsymbol{x}; t) = \mathbb{E}\left[1 + (\hat{o}_p(\boldsymbol{x}) - 1)\gamma\right] = t + (1 - t)\hat{o}_p(\boldsymbol{x}), \forall p \in [n]$$

Note that $0 \leq \hat{o}_p(\boldsymbol{x}) \leq 1$, we obtain $t \leq \hat{o}'_p(\boldsymbol{x}; t) \leq 1$. By using Assumption 1, we have

$$r(\boldsymbol{x})o_p(\boldsymbol{x}) \leq \hat{r}(\boldsymbol{x}) \leq \min\left\{\frac{1}{t}r(\boldsymbol{x})o_p(\boldsymbol{x}), 1\right\}. \tag{5}$$

This shows that $t$ can reduce the upper bound of $\hat{r}(\boldsymbol{x})$. Therefore, we obtain:

$$\exists p \in [n], \text{s.t.}, \hat{r}(\boldsymbol{x_1}) \leq \frac{1}{t}r(\boldsymbol{x_1})o_p(\boldsymbol{x_1}) \leq r(\boldsymbol{x_2})o_p(\boldsymbol{x_2}) \leq \hat{r}(\boldsymbol{x_2}),$$

where the first and the last inequality uses the Eq.(5), and the second inequality uses the Eq.(4).

### B.4 Proof of Theorem 4

From Eq.(5), we obtain:

$$\hat{r}(\boldsymbol{x}) \geq \sup_{p \in [n]} \{r(\boldsymbol{x})o_p(\boldsymbol{x})\} = r(\boldsymbol{x}) \sup_{p \in [n]} o_p(\boldsymbol{x}),$$

$$\hat{r}(\boldsymbol{x}) \leq \inf_{p \in [n]} \{\frac{1}{t}r(\boldsymbol{x})o_p(\boldsymbol{x})\} = \frac{1}{t}r(\boldsymbol{x}) \inf_{p \in [n]} o_p(\boldsymbol{x}).$$

Therefore we have: $r(\boldsymbol{x}) \sup_{p \in [n]} o_p(\boldsymbol{x}) \leq \frac{1}{t}r(\boldsymbol{x}) \inf_{p \in [n]} o_p(\boldsymbol{x})$, which implies the desired bound.

## C Experiments

### C.1 Further details of datasets

Table 2 shows the characteristics of the two datasets we used, Yahoo! and Istella-S.

Table 2: Dataset statistics

|  | Yahoo! | Istella-S |
|---|---|---|
| queries | 28,719 | 32,968 |
| documents | 700,153 | 3,406,167 |
| features | 700 | 220 |
| relevance levels | 5 | 5 |

### C.2 Training details

We combined the baselines with the same ranking models: DNN and Linear, where the implementation and hyperparameters are the same as ULTRA framework [5, 6]. To make fair comparisons, all the baselines and our models shared the same number of the hidden layer. The only difference between our model and baselines was the output dimensions. We trained these methods with a batch size of 256. We used SGD to train the Linear Ranker, and AdaGrad to train the DNN Ranker. For LBD, we selected the hyper-parameter $\lambda$ from $\{1, 100, 10000\}$ and $t$ from $\{0.1, 0.2, 0.3, 0.4\}$. For convenience, we expanded the output dimension of

Table 3: Final hyper-parameters used for LBD in all experiment settings

| Experiment setting | | | Final hyperparameter | | |
|---|---|---|---|---|---|
| Dataset | Ranker | $\eta$ | $\lambda$ | $t$ |
| Yahoo! | DNN | 0 | 100 | 0.1 |
| | | 0.1 | 100 | 0.1 |
| | | 0.2 | 100 | 0.1 |
| | | 0.3 | 1 | 0.1 |
| | | 0.4 | 1 | 0.3 |
| | | 0.5 | 1 | 0.3 |
| | | 0.6 | 1 | 0.4 |
| | Linear | 0.1 | 10000 | 0.1 |
| Istella-S | DNN | 0.1 | 100 | 0.1 |
| | Linear | 0.1 | 100 | 0.4 |

the ranking model, and separate them into the ranking model (with 2 outputs representing the mean and the variance) and observation models (with $2n$ outputs representing each observation distribution on each positions) as our implementation, since they have the same input data.

We used nDCG@$k$ ($k = 1, 3, 5, 10$) and ARP as the performance metrics. Each model was trained for 10,000 epochs, and we adopted the hyperparameters with the best results based on nDCG@10 tested on the validation set. We run each experiment 5 times and reported the average results testing on the test set. Final hyper-parameters used in all experiment settings are listed in Table 3, and the full experimental results are listed in Table 4.

Table 4: Comparison of different methods on two datasets ($\eta = 0.1$). Significant performance improvement/degradation compared to the best baseline (marked in bold) by t-test is denoted as +/- (p-value $< 0.05$) or ++/-- (p-value $< 0.005$).

| | | | nDCG@$k$ | | | |
|---|---|---|---|---|---|---|
| Ranker | Method | ARP | $k = 1$ | $k = 3$ | $k = 5$ | $k = 10$ |
| | | | Yahoo! | | | |
| DNN | Labeled Data | 3.536 | 0.692 | 0.696 | 0.716 | 0.764 |
| | **DLA** | 3.579 | 0.679 | 0.686 | 0.708 | 0.755 |
| | Vectorization | 3.593 | 0.674 | 0.681 | 0.704 | 0.751 |
| | PairDebias | 3.620 | 0.654 | 0.663 | 0.688 | 0.739 |
| | RegressionEM | 3.627 | 0.669 | 0.676 | 0.697 | 0.746 |
| | Click Data | 3.616 | 0.657 | 0.665 | 0.690 | 0.741 |
| | Unlimited | 3.585 | $0.671^-$ | $0.680^-$ | $0.703^-$ | $0.752^-$ |
| | LBD$_{Lips}$ | 3.572 | 0.680 | $0.689^+$ | $0.710^+$ | $0.757^{++}$ |
| | LBD$_{Ber}$ | 3.576 | 0.676 | 0.684 | $0.706^-$ | 0.754 |
| | LBD | **3.563$^+$** | **0.683$^+$** | **0.690$^{++}$** | **0.712$^{++}$** | **0.758$^{++}$** |
| Linear | Labeled Data | 3.603 | 0.671 | 0.676 | 0.698 | 0.747 |
| | DLA | 3.623 | 0.660 | 0.666 | 0.690 | 0.741 |
| | **Vectorization** | 3.633 | 0.663 | 0.667 | 0.690 | 0.741 |
| | PairDebias | 3.631 | 0.653 | 0.662 | 0.686 | 0.738 |
| | RegressionEM | 3.695 | 0.627 | 0.639 | 0.666 | 0.720 |
| | Click Data | 3.629 | 0.650 | 0.661 | 0.686 | 0.737 |
| | Unlimited | $3.710^{--}$ | $0.630^{--}$ | $0.640^{--}$ | $0.665^{--}$ | $0.719^{--}$ |
| | LBD$_{Lips}$ | $3.623^+$ | $0.669^+$ | $0.672^{++}$ | $0.695^{++}$ | $0.744^{++}$ |
| | LBD$_{Ber}$ | $3.623^+$ | 0.662 | 0.668 | 0.691 | 0.742 |
| | LBD | **3.616$^{++}$** | **0.671$^+$** | **0.674$^{++}$** | **0.696$^{++}$** | **0.746$^{++}$** |
| | | | Istella-S | | | |
| DNN | Labeled Data | 1.386 | 0.671 | 0.689 | 0.666 | 0.729 |
| | DLA | 1.551 | 0.642 | 0.613 | 0.637 | 0.696 |
| | Vectorization | 1.573 | 0.637 | 0.609 | 0.632 | 0.691 |
| | PairDebias | 1.496 | 0.618 | 0.596 | 0.624 | 0.691 |
| | RegressionEM | 2.064 | 0.592 | 0.556 | 0.573 | 0.623 |
| | **Click Data** | 1.475 | 0.636 | 0.610 | 0.636 | 0.701 |
| | Unlimited | $1.611^{--}$ | 0.635 | 0.606 | 0.630 | $0.687^{--}$ |
| | LBD$_{Lips}$ | $1.631^{--}$ | $0.640^-$ | 0.610 | $0.631^-$ | $0.686^{--}$ |
| | LBD$_{Ber}$ | **1.464$^+$** | $0.645^{++}$ | $0.618^{++}$ | $0.643^{++}$ | $0.707^{++}$ |
| | LBD | 1.469 | **0.651$^{++}$** | **0.623$^{++}$** | **0.648$^{++}$** | **0.709$^{++}$** |
| Linear | Labeled Data | 1.560 | 0.629 | 0.602 | 0.627 | 0.689 |
| | DLA | 1.687 | 0.612 | 0.584 | 0.609 | 0.667 |
| | Vectorization | 1.637 | 0.611 | 0.587 | 0.612 | 0.672 |
| | PairDebias | **1.554** | 0.604 | 0.582 | 0.613 | 0.678 |
| | RegressionEM | 2.214 | 0.552 | 0.522 | 0.542 | 0.594 |
| | **Click Data** | 1.564 | 0.610 | 0.588 | 0.616 | 0.680 |
| | Unlimited | $2.121^-$ | 0.542 | $0.514^-$ | $0.536^-$ | $0.592^-$ |
| | LBD$_{Lips}$ | $1.761^{--}$ | **0.619$^+$** | $0.583^-$ | $0.605^{--}$ | $0.659^{--}$ |
| | LBD$_{Ber}$ | $1.586^{--}$ | $0.613^+$ | 0.590 | 0.617 | 0.679 |
| | LBD | $1.581^{--}$ | $0.616^{++}$ | **0.591$^{++}$** | **0.618$^{++}$** | **0.680** |