# OpenReview forum: "LBD: Decouple Relevance and Observation for Individual-Level Unbiased Learning to Rank"
_NeurIPS.cc/2022/Conference — NeurIPS 2022 Accept_

### Official Review · Reviewer_FSFX · 2022-07-06

**Rating:** 4
**Confidence:** 4
**Soundness:** 2 fair
**Presentation:** 3 good
**Contribution:** 2 fair

**Summary:**

This paper studies the unbiased learning to rank problem. Different from existing work, the paper first argues that the bias factor should use any existing features, thus can go to individual-level. Then the authors argue that there will be a coupling effect between the relevance tower and bias tower as they share a lot of input signals. Then two simple methods are proposed. The first one enforces a smoothness constraint on the bias model (the so-called “Lipschitz Observation Model”). The second one uses dropout on the the bias model output (the so-called “Bernoulli Decoupling Model”). The paper provides some theoretical analysis on when decoupling is achievable. Experiments are done on semi-synthetic datasets with synthetic generated clicks on public datasets.

Overall the paper can be strengthened with more compelling theoretical analysis and experimental evaluation.

The reviewer acknowledges the author's reply. The dropout argument is decent. As some of the major concerns are still there (strong assumptions, only on semi-synthetic dataset, smoothness method only works on one dataset (more concerning as the datasets are manually controlled)). Will keep the score as the reviewer was initially between 3 and 4.

**Questions:**

The paper describes HTE as baseline in the text but it’s not shown in the results?
On Istella, “Click Data” can outperform several unbiased methods?


**Strengths And Weaknesses:**

Strength
The problem studied is important and the motivation is solid.
The theoretical analysis is interesting, though the assumption is strong and the usefulness is unclear.

Weakness
The methods are simple (which is good), but not novel. In fact, the dropout method was proposed 3 years ago (“recommending what video to watch next: a multitask ranking system”) but it was not cited. The smoothness method has several concerns (see below) and do not seem to perform well in experiments.
The theoretical analysis is not very compelling to the reviewer. The assumption is strong (the authors acknowledged it which is good). For example, the smoothness constraint is not very convincing - the authors already mentioned the position-only case. Also, it is intuitive that different query types can have quite different browsing behaviors, easily violating the smoothness constraint. It only shows feasibility but no guarantees. In practice it’s about hyper-parameter tuning.
The experiments are not satisfactory. First, using semi-synthetic dataset is common in the literature, so it does not add extra points (while many people use real-world datasets). Also, the reviewer feels it is more important to test this paper on more realistic datasets. Indeed, as the authors' motivation states, the bias can have complex correlation with input features. Existing work mainly generate bias only depends on position so it could be quite simple, but this work has a different motivation and the way it generates bias can be too simplified. Third, the authors are suggested to compare with some more recent methods (the most recent baseline was in 2019).

---

> ### Author Response · Authors · 2022-08-02
> **Response to Reviewer FSFX**
>
> Thank you for your detailed review and suggestions for improvements, we are glad that you enjoyed reading the paper. Below are our responses to the issues raised:
>
> **Concern 1: The dropout method is not novel.**
>
> We thank the reviewer for sharing this paper (and we will cite this paper in Section 4.3 in the final version), but we believe that our method is novel compare to it. Their work is different from our work in three folds: (1) The research scenarios are different: their work explicitly divided features into a relevance-related part and observation-related part (e.g., position), thus the coupling effect is naturally limited in their work. Our work considers a new coupling effect problem. (2) The motivation is different: The drop-out method in their work is used to bridge the gap between training and serving since the serving stage lacks position. Our dropout method is used to alleviate the couple effect problem, and we provide some theoretical analysis on it. (3) The implementation is different: Their dropout method takes effect on the input of the observation model, while our method takes effect on the output.
>
> **Concern 2: The smoothness method does not seem to perform well in experiments.**
>
> In the experiment, the smoothness method (LBD_Lips) performs significantly well (p-value < 0.005) in the Yahoo! dataset (eta = 0.1). We appreciate it if you could give more details on it.
>
> **Concern 3: The smoothness assumption is strong.**
>
> Currently, the widely-used assumption PBM is a special case of our Assumption 2 (note that we don't regard the position as a relevance-related feature), which assumes the Lipschitz constant is 0. So our assumption is a natural extension of the position-only case and can be applied to previous scenarios. Compared to traditional ULTR methods, we have made nontrivial progress to solve the smoothness case. We admit that the smoothness assumption may not be satisfied in some scenarios. However, the unsmoothness case is rather difficult to cope with and beyond the scope of this article. Here we give two possible future directions for improvement: (1) Inspired by [i], unlike the highly sparse high-dimensional original features, what really affects relevance and observation is a low-dimensional latent embedding. The effects of this latent embedding to observation can be seen as smooth. Thus, we may be able to learn this latent representation and then perform decoupling on it. (2) For some intractable features like query types, we can manually separate them into observation-related features and relevance-related features if possible. After manually removing the most obvious unsmooth factors, the remaining features that may affect observation can be seen as not obvious and smooth, which could be solved by our methods.
>
> [i] Counterfactual Prediction for Bundle Treatment, NeurIPS 2020
>
> **Concern 4: test on more realistic datasets**
>
> Before the paper is submitted, to the best of our knowledge, the only public ULTR dataset that contains both user click data and human-annotated relevance judgments is TianGong-ST [6]. However, as [6] reports, the features in this dataset are highly limited: only 33 simple text-matching features with limited expressive power, which is much less than the production system of the commercial Web search engine that has hundreds of features. Additionally, these features are unlikely to affect the user's observation greatly. We found empirically that our methods and baselines perform similarly on this dataset as well. Thus, we turn to do an experiment on the semi-synthetic dataset, which has hundreds of features from a real-world web search engine and is more closed to the production system. It's interesting to test our method on real-world click data, and we hope there is a proper public dataset in the future.
>
> **Concern 5: the way to generate bias can be too simplified.**
>
> Unlike existing work, our process of generating bias doesn't only simply depend on position but also on features. As mentioned in 6.1, we simulated the observation probability in a nonlinear fashion, which increases the difficulty of decoupling.
>
> **Concern 6: compare with some more recent methods**
>
> We have done the experiment to compare with the latest baseline: Vectorization (KDD 2022), and revised Table 1 and 3 in the paper. We have surveyed the recent related work but didn't find too many related papers. We are happy to incorporate more if the Reviewer has more suggestions on the baselines.
>
> **Concern 7: HTE is not shown in the results**
>
> We compared HTE in Figure 2(c) since it's not a jointly learning algorithm and cannot be included in Table 1.
>
> **Concern 8: “Click Data” can outperform several unbiased methods?**
>
> This is because these methods are based on position bias assumption, which may be away from the assumption we used and cannot correctly debias data. The phenomenon that Click Data can outperform some misspecified methods was also reported by other related work [6,14].

---

### Official Review · Reviewer_wGHG · 2022-07-06

**Rating:** 6
**Confidence:** 3
**Soundness:** 3 good
**Presentation:** 3 good
**Contribution:** 3 good

**Summary:**

In this work, the authors studied and proposed a Lipschitz and Bernoulli Decoupling (LBD) model to decouple relevance and observation at individual document level in unbiased learning-to-rank. The authors first explained about the coupling of relevance and observation, as there are common factors affecting both. They then translated the coupling effect into the main assumption of unbiased click prediction and defined both hard decoupling (model decomposes as the ground truth) and soft decoupling (model decomposition preserves the order of relevance function) based on that. Assuming the Lipschitz continuity of the observation function, the authors proved that the soft decoupling could be achieved by enforcing a Lipschitz continuity on the observation model with a finite Lipschitz constant and gave the upper and lower bound. Based on the same continuity assumption, the authors also showed Bernoulli observation model (BOM) can achieve the soft decoupling within the bounds of the observation dropout rate. Finally, the authors described the objective that combines both Lipschitz and Bernoulli Decoupling methods. In the experiment, the authors showed on the synthetic dataset, LBD could achieve the best unbiased results and also did the ablation studies on Lipschitz and Bernoulli alone. With a brief discussion of limitations, the authors concluded the paper. In the Appendix, the authors extended related works, proved the main theorems, and included more experiment details.

**Questions:**

Good that authors discussed the limitations on Assumption 1. On the other hand, based on my understanding, both Lipschitz and Bernoulli methods are based on the Assumption 2, i.e., the Lipschitz continuity assumption of observation. In general, I tend to accept the assumption. However, on the other hand, it implies the function is smooth up to certain scale. If we need to learn the smooth function and extend to below the scale, we need in our dataset at least dense enough to cover to the scale, especially, when our observation is now in a much higher dimension space. Can authors discuss this limitation on Assumption 2 as well? And explain the requirements on the dataset, e.g. data size, and relation to the smoothness alpha?

Please also check my other questions in weakness.

**Limitations:**

Please see my comments in questions and weakness above.

**Strengths And Weaknesses:**

Strengths:
1. The work studies a general and important but previously ignored issue in unbiased learning to rank, which has been the core of industrial application in recent years.
2. Well written and solid theory.
3. Complete study with sufficient ablation study and discussion on limitations.

Weaknesses:
1. The authors only demonstrated their method on the synthetic dataset, despite the method is claimed to be applicable to more nontrivial bias. It could be more convincing if authors could test the method on real-world click data.
2. The authors should better double-check their significance analysis. There are some results appearing absurd: e.g. in Table 1, 0.659 of LBD_Lips Linear model on Istella-S is significant worse while 0.592 of Unlimited Linear model is not significant? Also, not sure which exactly the baseline was when applied the significance analysis, why not 0.686 and 0.687 are not significant if 0.707 is significant, assuming the baseline is the best one with 0.701.
3. In Figure 2(c), when authors claimed about better performance of LBD at low eta. In fact, they only have a single point LBD performs better, which is eta=0, exactly no coupling. Isn't that somewhat deviated from the main claim of the paper. Can authors add some points, for example, 0<eta<0.1, to show LBD does work better at low eta end?
4. Minor points on notations: a. variable t appears to be doubly used in Theorem 2 and in Bernoulli model parameter; b. Appendix chapters should better always be clarified, somewhat confused when not knowing there is an appendix.

---

> ### Author Response · Authors · 2022-08-02
> **Response to Reviewer wGHG**
>
> Thank you for the valuable feedback!
>
> **Concern 1: Experiments on real world click data.**
>
> Before the paper is submitted, to the best of our knowledge, the only public ULTR dataset that contains both user click data and human-annotated relevance judgments is TianGong-ST [6]. However, as [6] reports, the features in this dataset are highly limited: only 33 simple text-matching features with limited expressive power, which is much less than the production system of the commercial Web search engine that has hundreds of features. Additionally, these features are unlikely to affect the user's observation greatly. We found empirically that our methods and baselines perform similarly on this dataset as well. Thus, we turn to do an experiment on the semi-synthetic dataset, which has hundreds of features from a real-world web search engine and is more closed to the production system. It's interesting to test our method on real-world click data, and we hope there is a proper public dataset in the future.
>
> **Concern 2: Significance analysis**
>
> We thank the Reviewer for pointing out the problem of 0.686 and 0.687 (both of them are significant). Besides, some methods have large standard deviations (0.592 of Unlimited), which increases the p-value (0.0291<0.005). So the 0.592 is not significant.
>
> **Concern 3: Add more points on $0<\eta<0.1$**
>
> We have done these experiments (add points on $\eta=0.01$ and $\eta=0.05$):
>
> |$\eta$|LBD|HTE-Oracle|
> |-|-|-|
> ||nDCG@10|nDCG@10
> |$0.00$|$0.7604\pm0.0008\$|$0.7585\pm0.0006$
> |$0.01$|$0.7601\pm0.0009$|$0.7592\pm0.0005$
> |$0.05$|$0.7602\pm0.0006$|$0.7595\pm0.0011$
> |$0.10$|$0.7587\pm0.0011$|$0.7602\pm0.0011$
>
> and we revised Figure 2(c). Besides, it's worth mentioning that the fair comparison for LBD is HTE instead of HTE-oracle, since HTE-oracle uses additional information and can only serve as an upper bound of HTE. We can see that when $0<\eta<0.2$, our method is better than HTE, which is consistent with the main claim of our paper.
>
> **Concern 4: Minor points**
>
> - a) doubly used: Thanks for spotting the problem, we have fixed it in the revision.
> - b) appendix: In the main content, we have provided a pointer to the Appendix to achieve self-containing. If you could give more suggestions to make our article more readable, we can improve it.
>
> **Concern 5: Limitation on Assumption 2**
>
> Thanks for pointing out the limitation of Assumption 2. Our bounds in the theorems contain an $\alpha$ from Assumption 2, and it is related to the real data generating process. In some cases such as sparse, high dimensional, or high noise cases, the value of $\alpha$ will be too large and invalidate the derived bounds. We also provide a possible future direction for improvement, and you can refer to our response to Concern #3 from R FSFX.

---

### Official Review · Reviewer_jZaM · 2022-07-11

**Rating:** 6
**Confidence:** 3
**Soundness:** 2 fair
**Presentation:** 3 good
**Contribution:** 3 good

**Summary:**

The authors propose an Unbiased Learning-to-rank (ULTR) method which relies on only clicks and performs decoupling to separate relevance and observation parts from the click model.

Given that we observe only clicks, it is very difficult to retrieve the true relevance and observation parameters, because multiple combinations of relevance and observation can result in the same click.

The authors propose a disentangling method to decouple the two parameters, with theoretical guarantees.

Empirical results on the standard semi-synthetic benchmark demonstrate the efficacy of the method over established baselines.

-- I have read and considered the author's response.

**Questions:**

I couldn't find the proof for soft decoupling and hard decoupling in the draft. It points to an Appendix, but I couldn't find one.


**Strengths And Weaknesses:**

Some strong points of the paper:

- The paper is well-written, with a proper introduction to the area of ULTR and biases in click data. Given the limited space in the NeuRIPS format, the authors have done a commendable job of introducing the related works and the problem itself.
- The assumptions are laid out in an intuitive manner and the method is supported by theory.
- Empirical results indicate strong performance as compared to baselines.

Some weak points/comments on the draft:
- During training, when clicks are estimated via a product of predicted relevance and predicted observations, even with the regularization term on the norm of gradient, a trivial solution would be to make the observation model's prediction constant, and relevance equal to clicks. How will the model recover the true relevance in such a case? (even with soft decoupling, it needs to estimate relevance up to a constant).

Overall, I think the paper presents an interesting theoretically motivated algorithm for ULTR.

---

> ### Author Response · Authors · 2022-08-02
> **Response to Reviewer jZaM**
>
> **Concern 1: How will the model recover the true relevance when the observation model's prediction is constant?**
>
> We have discussed it in Theorem 2. This trivial solution (i.e., LOM with $\beta$ = 0) will cause the model to not be able to fit the click rate, which violates Assumption 1. Thus, the norm of the gradient cannot be unlimited small.
>
> **Concern 2: The proofs for decouplings are missing.**
>
> Soft decoupling and hard decoupling are definitions, rather than theorems. These two definitions describe the targets of our work, and Theorems 1-4 (proofs can be found in the Appendix) describe methods to achieve the targets.

---

> > ### Comment · Reviewer_jZaM · 2022-08-07
> > **Thanks for the response.**
> >
> > Thank you authors for your response.
> >
> > I acknowledge the proof in the appendix.
> >
> > Thanks again.

---

### Official Review · Reviewer_7htM · 2022-07-11

**Rating:** 6
**Confidence:** 4
**Soundness:** 2 fair
**Presentation:** 3 good
**Contribution:** 4 excellent

**Summary:**

The authors propose a debiasing method in the learning to rank (L2R) regime. They advance the hypothesis that contemporary debiasing methods are lacking since they assume a click model where the effect of the position of an item in a ranked list on the probability of observation by the user is constant across different contexts.

The authors instead propose learning parametric models for observation probability and relevance probability. Since only clicks are observed in practice and they assume P(click|X) = P(observation|X, position) * P(relevance|X), the two quantities of observation/relevance are not identifiable. As a result the authors instead propose two conditions to fulfill a "soft decoupling" condition that maintains ranks in their observational model when compared to the true probabilities of observation.
1. lipschitz decoupling: where P(observation|X, position) is alpha-lipschitz in X and its corresponding estimation model is also beta-lipschitz in X, for a value of beta that falls within a bound dependent on alpha.
2. bernoulli decoupling: the observation probability is once again alpha-lipschitz, but here the corresponding estimation model is instead forced to output 1 with probability t, forcing the ranker to learn directly from click data. If t falls within a bound dependent on alpha, then soft decoupling also applies here.

The authors then validate their model across a linear and neural network ranker and compare to a number of unbiased baselines in semi-synthetic L2R settings.

**Questions:**

1. Why is the bernoulli model the right model here? While I read the paper the choice felt a bit arbitrary, and rather felt like a hack to enforce soft coupling, rather than reflecting any real data generating process.
2. Does soft decoupling give us any guarantees about ranking accuracy? For example, if I have a softly decoupled observational model with probabilities that are way off from the true probabilities, can I still have some guarantee about my click probability?
3. There seems to be a gap between the theoretical and empirical sections of the paper since you tune the beta and t hyperparameters using a validation set. Given that you have access to the true underlying probabilities in the simulation environment, what happens if you explicitly compute the lipschitz bound
4. I'm confused how you evaluated nDCG in your validation set and test set. I had assumed you only observed click data across all splits of the data, hence shouldn't all your items in each query have a score of 0 except for the clicked item? In that case, wouldn't it be more interpretable to use a metric such as the average position of the clicked item, or the mean reciprocal rank?

**Limitations:**

Given that the intent of L2R models is for them to be deployed on large internet platforms, I would have liked to have seen some discussion about the societal impact of the work.

**Strengths And Weaknesses:**

This is a good contribution to the area of L2R. The model they study is elegant and natural and I was surprised there has been no prior work making similar assumptions.

Overall the clarity of exposition was very good and the paper was easy to follow. However, I felt that the authors could have provided more intuition in the bernoulli decoupling section (see questions).

While the ideas presented are quite valuable I felt the results in the experiments were a bit underwhelming. The authors assume that their hypothesis about disparate effects of rankings is correct in their simulations and only show marginal improvements in nDCG when compared to DLA unless their coupling parameter is significantly increased (the nDCG difference between DLA and LBD are smaller than the differences between LBD and itself for two different values of the hyperparameter t that are apart by 0.1). However, I did appreciate the completeness of their experimental results: the authors included results across a wide range of simulation parameters and compared their method with a broad range of baselines.

Overall, I'm leaning towards an accept although I would be willing to change my score up from a weak accept to an accept if the authors are able to make a compelling case for the strength of their experimental results and address my questions below.

---

> ### Author Response · Authors · 2022-08-02
> **Response to Reviewer 7htM**
>
> Thank you for your valuable comments and feedback. Below are our responses to the issues raised:
>
> **Concern 1: Experiment Strength**
>
> Our method is similar to DLA (a similar listwise loss function). Since DLA is designed to deal with no-coupling cases, the two models should have performed similarly when the coupling effect is weak. Empirically, we find that when the coupling parameter is low (0-0.1), our method won't degrade and even slightly better than DLA. With an increased coupling parameter, the gap between DLA and our method is larger, which is able to verify our method in decoupling relevance and observation.
>
> **Concern 2: Why is the Bernoulli model the right model?**
>
> This involves how to define the "right model". In our work, we claim that if a model can achieve soft decoupling (no need to recover the real data generating process), then it is right in the unbiased LTR task. We give an intuition explanation of the Bernoulli model as follows: Since the features are generally collected for the ranking task, the effects from features to relevance are usually stronger than the ones from features to observation. Thus, we hope the relevance model could learn more effects from features to clicks than the observation model. When Lipschitz Decoupling fails, the Bernoulli sampling prevents the relevance model from over-relying on the wrong observation model and encourages the relevance model to relearn what was learned by the observation model. The probability $t$ can be seen as a global parameter to control the strength of relevance effects.
>
> **Concern 3: Does soft decoupling give us any guarantees about ranking accuracy?**
>
> The soft decoupling indicates that the ranking order must be correct, even if the observation model's prediction is way off from the true probabilities. For example, if an observation model always predicts twice the true observed probability (which differs from the true probability) and the click probability can be estimated correctly (Assumption 1), then the relevance probability is exactly half the true relevance probability, which also achieves soft decoupling since the ranking order remains the same. Besides, the definition of soft decoupling is based on Assumption 1, so if the models could soft decouple, the click probability estimation should be correct.
>
> **Concern 4: A gap between the theoretical and empirical sections.**
>
> The bounds are difficult to compute explicitly for two reasons: (1) the $\beta$ in the bounds of Lipschitz Decoupling have no closed-form solution. (2) the bounds focus on the pairwise comparison of documents. Even though we can theoretically find the bounds for individual documents, it's difficult to find the best $\beta$ and $t$ for the global dataset. So we still need to tune the hyperparameters for the whole dataset. This is similar to [i]: The proposed objective function is inspired by the derived bound, and the Lipschitz constants in the bounds are treated as hyperparameters, which are tuned on the validation dataset.
>
> [i] Kai Yu et al., Nonlinear Learning using Local Coordinate Coding.
>
> **Concern 5: How is nDCG evaluated in the validation set and test set?**
>
> Our validation set and test set contain the true labels (5 levels), which follow the experimental specification of the existing ULTR [4, 5, 14, 21]. Besides, except nDCG, we also calculate the ARP metrics (assuming that the documents with label = 0 are irrelevant and the other documents are relevant) and show the results in Tabel 3, Appendix.
>
> **Concern 6: Societal impact**
>
> To the best of our knowledge, the approaches in this paper raise no major ethical concerns and societal consequences. Researchers and practitioners from the ULTR domain may benefit from our research since debiasing implicit feedback data is a significant challenge in real-world applications. The worst possible outcome when the proposed approach fails is that it reduces to the standard position-based observation estimation and stops making the desired impact. Finally, the proposed approach aims at solving the coupling effects of the data, the extent of which depends on the properties of the data.

---

### Author Response · Authors · 2022-08-02
**General Response to all Reviewers**

We thank the Reviewers for insightful comments and valuable feedback. We are delighted that they found our work to be well-motivated (R jZaM, R FSFX), the problem studied is highly novel and important in the area of LTR (R 7htM, R wGHG, R FSFX), and the theoretical analysis is solid and interesting (R jZaM, R wGHG, R FSFX). They agree that our paper is well-written (R 7htM, R jZaM, R wGHG), and the experiments are sufficient (R 7htM, R wGHG) and indicate strong performance (R jZaM). We appreciate the Reviewers’ valuable suggestions and will incorporate all feedback into the final version of our work. In the following, we summarize key changes in the revision in response to the Reviewers’ concerns.

- We add an intuitive explanation of Bernoulli Decoupling in section 4.3.
- We add more discussion on the limitation of Assumption 2, and add a Societal Impact session.
- We test the latest baseline, Vectorization ("Scalar is Not Enough: Vectorization-based Unbiased Learning to Rank", KDD 2022) on our setting, and revise Table 1 and 3 (Thank R wGHG for pointing out the problem of significant analysis).
- We add more points from 0 to 0.1 in Figure 2(c).

In the following, we will address the concern of each Reviewer individually.

---

### Author Response · Authors · 2022-08-07
**Anything else the Reviewers would like us to respond to?**

Dear Reviewers,

We would like to check with the Reviewers if our replies have addressed your concerns. Additionally, if you still have any further questions, please let us know and we'd be happy to take a look. Thank you.

---

### Meta-Review · Area_Chair_ETcC · 2022-08-26

**Recommendation:** Accept
**Confidence:** Less certain

**Metareview:**

The paper studies the unbiased learning to rank problem, and introduces new assumptions and techniques to learn good ranking policies from biased data. The key insight is to use smoothness assumptions to decouple the effect of observation from relevance. The reviewers appreciated this novel attack on a significant problem, and the authors clarified how existing position-based models and other debiasing strategies emerge as special cases of their smoothness assumptions.

Some reviewers pointed out deficiencies in the empirical study -- lack of results on real-world click data, questions around statistical significance and finer-grained sweeps of hyper-parameter ranges (which the authors subsequently clarified during the feedback phase). Adding a real-world experiment, even on the limited TianGong-ST dataset that the authors identified but rejected, would substantially strengthen the claims in the paper. That said, the Lipschitz and Bernoulli decoupling are likely to be of interest to the learning to rank community, and spur follow-up work on better user modeling.

**Award:**

No

---

### Decision · Program_Chairs · 2022-09-14

Accept